# A Comparison of Abstraction Heuristics for Rubik's Cube

**Clemens Büchner**[1], **Patrick Ferber**[1,2], **Jendrik Seipp**[3], **Malte Helmert**[1]

[1]University of Basel, Switzerland
[2]Saarland University, Saarland Informatics Campus, Saarbrücken, Germany
[3]Linköping University, Sweden
{clemens.buechner, patrick.ferber, malte.helmert}@unibas.ch, jendrik.seipp@liu.se

## Abstract

Since its invention in 1974, the Rubik's Cube puzzle fascinates people of all ages. Its rules are simple: the player gets a scrambled cube and rotates the six faces until each face contains only stickers of one color. Nevertheless, finding a short sequence of rotations to solve the cube is hard. We present the first model of Rubik's Cube for general problem solvers. To obtain a concise model, we require conditional effects. Furthermore, we extend counterexample-guided Cartesian abstraction refinement (CEGAR) to support *factored effect tasks*, a class of planning tasks with a specific kind of conditional effects which includes Rubik's Cube. Finally, we evaluate how newer types of abstraction heuristics compare against pattern database (PDB) heuristics, the state-of-the-art for solving Rubik's Cube. We find that PDBs still outperform the more general Cartesian and merge-and-shrink abstractions. However, in contrast to PDBs, Cartesian abstractions yield perfect heuristics up to a certain problem difficulty. These findings raise interesting questions for future research.

## Introduction

In *classical planning*, we aim to find (short) paths in large, deterministic transition systems. In general, this means that we search for a sequence of actions leading from the initial state of the problem to a state which satisfies some goal condition. *Rubik's Cube* is a difficult classical planning problem which also enjoys prominence in the general public. The traditional Rubik's Cube consists of 27 small cubes called *cubies* colored differently on each face. The cubies are stacked in the shape of a larger $3 \times 3 \times 3$ cube. In every dimension (horizontal, vertical, and depth), the cube has 3 layers which can be rotated independently. Rotating a layer changes the colors on the faces of Rubik's Cube. Given an arbitrarily rotated Rubik's Cube with colors scrambled all over its faces, the goal is to find a sequence of rotations which brings all faces to show a single color only.

Rubik's Cube has approximately $4.3 \cdot 10^{19}$ reachable states; too many to find solutions using blind search. To solve such hard classical planning tasks, *heuristic search* has proven a very successful method (e.g., Bonet and Geffner 2001; Hoffmann and Nebel 2001; Helmert and Domshlak 2009; Richter and Westphal 2010; Helmert et al. 2014; Domshlak, Hoffmann, and Katz 2015). A *heuristic* is a function that estimates the cost from a given state to the closest

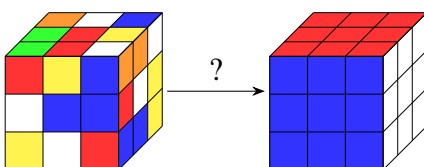

Figure 1: The classic $3 \times 3 \times 3$ Rubik's Cube, scrambled on the left and solved on the right.

goal state. In the case of Rubik's Cube, the cost to the goal is the number of rotations required to reach the goal. The guidance of a good heuristic helps the search to focus on states that are likely part of a shortest solution.

One family of heuristics are *abstraction heuristics* (Seipp and Helmert 2018). An abstraction is an equivalence relation between states. The states of the original problem within the same equivalence class are mapped to a single abstract state. Each transition in the original problem induces a transition between the corresponding abstract states in the abstraction. Since the abstract problem generally has fewer states but preserves transitions between concrete states, it is easier to solve than the original problem. Moreover, the cost of a shortest path between two concrete states is lower-bounded by the cost of a shortest path between the corresponding abstract states. Abstraction heuristics therefore use the abstract goal distance as an admissible estimate of the goal distance in the original problem.

The state-of-the-art for finding optimal solutions for Rubik's Cube are *pattern database* (PDB) heuristics (Korf 1997; Culberson and Schaeffer 1998). PDBs are abstraction heuristics and were developed in the 90's. Since then, *domain abstractions* (Hernádvölgyi and Holte 2000), *Cartesian abstractions* (Seipp and Helmert 2018), and *merge-and-shrink (M&S) abstractions* (Sievers and Helmert 2021) have been introduced to classical planning. These methods differ in the way they map states to abstract states and can be ordered by increasing generality: PDBs, domain abstractions, Cartesian abstractions, and M&S abstractions. This raises the question how the more general abstractions compare against the less general PDBs for Rubik's Cube.

Our contribution is three-fold. First, we show how Cartesian abstractions can be generated for *factored effect tasks*, a class of planning tasks where all operators exclusively have

conditional effects which are conditioned only on the variable they change. Second, we model Rubik's Cube in finite-domain representation (Helmert 2009). This enables us to use common general problem solvers on Rubik's Cube. To avoid the enormous number of ground operators in this problem, our model requires *conditional effects* which are the reason for the generalization of Cartesian CEGAR in the first place. Third, we compare the different abstraction heuristics against one another on Rubik's Cube tasks. We leave out domain abstractions in this analysis because we are not aware of any techniques to generate them in a principled way.

Our experimental evaluation reveals that PDBs are still the strongest technique for finding optimal solutions for Rubik's Cube, even though we are unable to use Korf's (1997) large patterns. Among all compared heuristics, the PDB heuristic based on our adaption of Korf's patterns still solves the most problems from our newly generated Rubik's Cube benchmarks. The heuristic accuracy, however, is surpassed by our adaption of the CEGAR algorithm for generating Cartesian abstractions (Seipp and Helmert 2018). M&S abstractions are the least successful for Rubik's Cube according to our findings. This is counter-intuitive, as one might expect that the more general abstraction classes should result in better problem solvers. We discuss why this might not be the case for our case study of Rubik's Cube towards the end of the paper.

## Background

**Classical Planning**   We consider classical planning tasks in finite-domain representation (Helmert 2009), an extension of the $SAS^+$ formalism (Bäckström and Nebel 1995) with *conditional effects*. A *planning task* is a 4-tuple $\Pi = \langle \mathcal{V}, \mathcal{O}, s_I, s_\star \rangle$. $\mathcal{V}$ is a finite set of variables. Each variable $V \in \mathcal{V}$ has a finite domain $\mathrm{dom}(V)$. We call a variable-value assignment $V \mapsto v$ with $V \in \mathcal{V}$ and $v \in \mathrm{dom}(V)$ an *atom*. A *partial state* $s$ is a function over a set of variables $vars(s) \subseteq \mathcal{V}$ and assigns each variable $V \in vars(s)$ a value $v = s[V] \in \mathrm{dom}(V)$. A partial state $s$ where $vars(s) = \mathcal{V}$ is a *state*. $S(\Pi)$ is the set of all states of $\Pi$. We sometimes treat (partial) states as sets of atoms. $\mathcal{O}$ is a finite set of *operators*. Each operator $o \in \mathcal{O}$ has a *precondition* $pre(o)$ which is a partial state, a finite set of *effects* $eff(o)$, and a *cost* $cost(o) \in \mathbb{R}_0^+$. Each effect is a triple $\langle c, V, v \rangle$ where $c$ is a partial, possibly empty, state called the *effect condition*, $V \in \mathcal{V}$ is a variable, and $v \in \mathrm{dom}(V)$ is the new value for $V$. An operator $o$ is *applicable* in state $s$ iff $pre(o) \subseteq s$. Applying an applicable operator $o$ in state $s$, called *progression* and written as $s[\![o]\!]$, leads to a *successor state* $s'$. For each variable $V \in \mathcal{V}$, $s'[V]$ is defined as

$$s'[V] = s[\![o]\!][V] = \begin{cases} v & \text{if } \exists \langle c, V, v \rangle \in eff(o) : c \subseteq s \\ s[V] & \text{otherwise.} \end{cases}$$
(1)

This definition assumes that effects are *conflict-free*, i.e., it is never ambiguous which value is assigned to a variable. More formally, for each state $s$ and each applicable operator $o$, there are no tuples $\langle c_1, V, v_1 \rangle, \langle c_2, V, v_2 \rangle \in eff(o)$ with $c_1 \subseteq s$ and $c_2 \subseteq s$ and $v_1 \neq v_2$. Finally, $s_I \in S(\Pi)$ is the *initial state* and $s_\star$ is a partial state called the *goal*.

We also need the notion of *regression*. While progression provides for a state $s$ and an operator $o$ the successor state $s[\![o]\!] = s'$, regression provides for a state $s'$ and an operator $o$ the set of predecessor states $regr(s', o)$ such that $s \in regr(s', o)$ iff $s[\![o]\!] = s'$.

An *operator sequence* $\pi = \langle o_1, \ldots, o_n \rangle$ is applicable in a state $s_0$ if every operator $o_i$ is applicable in $s_{i-1}$ and $s_i = s_{i-1}[\![o_i]\!]$ in turn. We write $s[\![\pi]\!]$ to denote the final state that results from applying $\pi$ in $s$. The *cost* of an operator sequence $\pi$ is the sum of the costs of its operators, i.e., $cost(\pi) = \sum_{i=1}^n cost(o_i)$. We call an operator sequence $\pi$ a *plan* if it is applicable in $s_I$ and $s_I[\![\pi]\!] \supseteq s_\star$. A plan $\pi$ is *optimal* if its cost is minimal among all plans.

A *transition system* $\mathcal{T} = \langle S, \mathcal{L}, T, s_I, S_\star \rangle$ is an edge-labeled, directed graph with nodes $S$, transition labels $\mathcal{L}$, transitions $T \subseteq S \times \mathcal{L} \times S$, initial node $s_I \in S$, and goal nodes $S_\star \subseteq S$. Every planning task $\Pi = \langle \mathcal{V}, \mathcal{O}, s_I, s_\star \rangle$ induces a transition system $\mathcal{T}(\Pi) = \langle S(\Pi), \mathcal{O}, T(\Pi), s_I, \{s \in S(\Pi) \mid s_\star \subseteq s\} \rangle$ where $T(\Pi) = \{\langle s, o, s' \rangle \mid s, s' \in S(\Pi), o \in \mathcal{O}, pre(o) \subseteq s, s' = s[\![o]\!]\}$.

An (optimal) plan for $\Pi$ corresponds to a (shortest) path from $s_I$ to its closest goal node in $\mathcal{T}(\Pi)$ and vice versa. Most algorithms today use progression to find (shortest) paths in $\mathcal{T}(\Pi)$. One such algorithm is A* (Hart, Nilsson, and Raphael 1968) which systematically explores the transition system and uses a *heuristic function* to estimate the cost to reach the goal for each node. A* in particular comes with the guarantee that if the heuristic is *admissible*, i.e., it never overestimates the cost to the goal, then the path it finds is optimal.

**Abstractions**   Let $\mathcal{T} = \langle S, \mathcal{L}, T, s_I, S_\star \rangle$ be a transition system. An *abstraction* $\alpha : S \to S^\alpha$ maps the states of $\mathcal{T}$ to a set of *abstract states* $S^\alpha$. The induced transition system is $\mathcal{T}^\alpha = \langle S^\alpha, \mathcal{L}, T^\alpha, \alpha(s_I), \{\alpha(s) \mid s \in S_\star\} \rangle$ where $T^\alpha = \{\langle \alpha(s), o, \alpha(s') \rangle \mid \langle s, o, s' \rangle \in T\}$. By construction, every path in $\mathcal{T}$ is a path in $\mathcal{T}^\alpha$. Consequently, the length of a shortest path between state $\alpha(s)$ and $\alpha(s')$ in $\mathcal{T}^\alpha$ is a lower bound on the length of a shortest path between state $s$ and $s'$ in $\mathcal{T}$. Thus, the abstract goal distance for a given state is an admissible estimate of the true goal distance.

There are three common abstraction heuristics: pattern databases (PDBs), Cartesian abstraction heuristics, and merge-and-shrink (M&S) heuristics.

PDBs are based on projections. The projection $\phi_P(s)$ of a (partial) state $s$ onto a set of variables $P \subseteq \mathcal{V}$ (called *pattern*) is $\{V \mapsto v \mid (V \mapsto v) \in s, V \in P\}$. $\phi_P$ is an abstraction and induces an abstract transition system $\mathcal{T}^{\phi_P}$. For every abstract state of $\mathcal{T}^{\phi_P}$, a PDB calculates and stores its distance to the goal. When queried for a state $s$, the PDB returns the stored value for $\phi_P(s)$.

Cartesian abstractions are a generalization of PDBs. Let $\mathcal{V} = \{V_1, \ldots, V_n\}$ be the set of variables and $A_i \subseteq \mathrm{dom}(V_i)$ be subsets of their domains. Then $A = A_1 \times \cdots \times A_n$ is a Cartesian set which represents the states $\{s \in S(\Pi) \mid s[V_i] \in A_i\}$. We define $A[V_i] = A_i$. An abstraction is called Cartesian if all its abstract states are Cartesian sets.

Cartesian abstractions can be obtained by using counterexample-guided Cartesian abstraction refinement (CEGAR, Seipp and Helmert 2018). CEGAR starts

with an abstract transition system $\mathcal{T}^\alpha$ with a single abstract state. Afterwards, it iteratively generates a plan $\pi^\alpha$ for $\mathcal{T}^\alpha$, executes $\pi^\alpha$ on the original task until it finds a flaw, and then refines $\mathcal{T}^\alpha$ by splitting the abstract state where $\pi^\alpha$ failed such that the flaw does not occur in future iterations. This refinement loop repeats until some stopping criterion (e.g., timeout) is reached. The final abstract transition system is used as heuristic like in the PDB setting. If at any point no plan exists, then $\mathcal{T}^\alpha$ is unsolvable and so is the original task. If at any point there is no flaw, then the plan is also a solution to the original task. For a transition $\langle A, o, B \rangle \in \pi^\alpha$ and the concrete state $s$, there are two kinds of flaws: (1) $o$ is not applicable in $s$ and (2) $s[\![o]\!] \notin B$. To refine the latter flaw, we require the regression of an operator and an abstract state, i.e., a set of abstract states $regr(B, o)$ from which we reach $B$ via the operator $o$. For more information regarding the refinement, we refer the reader to Seipp and Helmert (2018).

M&S (Sievers and Helmert 2021) produces even more general abstractions. The algorithm starts with a pool of abstract transition systems. Initially, this pool contains for every variable $V \in \mathcal{V}$ the abstract transition system $\mathcal{T}^{\pi_{\{V\}}}$ of the projection $\pi_{\{V\}}$. M&S iteratively merges two transition systems $\mathcal{T}^1$ and $\mathcal{T}^2$ from the pool by replacing them with their cross product $\mathcal{T}^\otimes$. If necessary, $\mathcal{T}^\otimes$ is shrunk by combining some of its states to reduce the memory footprint. These steps are repeated until exactly one abstract transition system is left in the pool. As for PDBs, the perfect heuristic is calculated and stored for every state $s$ of the final abstract transition system and used as heuristic for the state of the original transition system.

## Cartesian Abstraction Refinement for Factored Effect Tasks

Seipp and Helmert (2018) introduced counterexample-guided Cartesian abstraction refinement (CEGAR) for classical planning. Their theory is limited to planning tasks without conditional effects, i.e., they require $c = \emptyset$ for all effects $\langle c, V, v \rangle \in eff(o)$ for all operators $o \in \mathcal{O}$. In this section, we extend this theory to a more general kind of planning tasks that we call *factored effect tasks*.

### Factored Effect Tasks

The structure of our planning task model for Rubik's Cube (see next section) has a distinctive characteristic: every effect is conditioned only on the variable it changes. Besides Rubik's Cube, this characteristic is useful to model other permutation problems such as the *15-puzzle*, the *pancake problem*, *genome rearrangement*, and more. We define factored effect tasks to describe arbitrary planning tasks with this characteristic as follows.

**Definition 1.** *Factored Effect Operator.*
*An operator $o \in \mathcal{O}$ is a* factored effect operator *if every effect $e \in eff(o)$ is a triple $e = \langle \{V \mapsto v\}, V, v' \rangle$ for some $v, v' \in \mathrm{dom}(V)$ with $v \neq v'$. We abbreviate the notation to $e = \langle V, v, v' \rangle$.*

*We write $vars(eff(o)) = \{V \mid \langle V, v, v' \rangle \in eff(o)\}$ to denote all variables affected by $o$. Furthermore, we require*

*that all factored effect operators $o$ satisfy $v_1 \neq v_2$ for all pairs $\langle V, v_1, v_1' \rangle, \langle V, v_2, v_2' \rangle \in eff(o)$ affecting the same variable $V$ for all $V \in \mathcal{V}$, which guarantees that $o$ is conflict-free.*

**Definition 2.** *Factored Effect Task.*
*A* factored effect task *is a planning task $\Pi = \langle \mathcal{V}, \mathcal{O}, s_I, s_\star \rangle$ where all $o \in \mathcal{O}$ are factored effect operators.*

### Cartesian CEGAR

There are two obstacles to using the original Cartesian CE-GAR procedure (Seipp and Helmert 2018) in the presence of general conditional effects: (1) The regression of a Cartesian set with respect to an operator with conditional effects is not necessarily Cartesian (violation of Seipp and Helmert's Property 4); (2) It is impossible to infer which facts hold after applying an operator without a concrete state as context (required for Seipp and Helmert's Algorithm 5). We show in the following two theorems that these two obstacles disappear when considering factored effect tasks.

**Theorem 1.** *The regression of a Cartesian set with respect to a factored effect operator is Cartesian.*

*Proof.* Let $B = B_1 \times \cdots \times B_n$ be a Cartesian set representing an abstract state over variables $\mathcal{V} = \{V_1, \ldots, V_n\}$ where $B_i \subseteq \mathrm{dom}(V_i)$. A factored effect operator $o$ changes the value of a variable independent of the other variables. Hence, we can reason about all variables individually.

Let us consider the regression of the single variable $V_i$. There are two justifications for $b \in B_i$ after applying $o$ in an abstract state $A = A_1 \times \cdots \times A_n$: either $b \in A_i$ and there is no effect $\langle V_i, b, x \rangle \in eff(o)$ or $\langle V_i, a, b \rangle \in eff(o)$ where $a \in A_i$. This leads to the following definition of possible values $A_i'$ preceding $B_i$.

$$A_i' = \{b \in B_i \mid \nexists \langle V_i, b, x \rangle \in eff(o)\} \cup$$
$$\{a \qquad \mid \exists \langle V_i, a, b \rangle \in eff(o) : b \in B_i\}$$

Since we can only reach $B$ from states where $o$ is applicable, $A_i$ is restricted to values satisfying the precondition of $o$.

$$A_i = \begin{cases} A_i' \cap \{pre(o)[V_i]\} & \text{if } V_i \in vars(pre(o)) \\ A_i' & \text{otherwise} \end{cases}$$

Then, the regression of Cartesian set $B$ with respect to factored effect operator $o \in \mathcal{O}$ is $regr(B, o) = A_1 \times \cdots \times A_n$. Clearly, $A_i \subseteq \mathrm{dom}(V_i)$ and thus $regr(B, o)$ is Cartesian. $\square$

To address (2), we redefine the *post*-function of Seipp and Helmert (2018). The *post*-function describes a partial state which holds after applying an operator $o$. In the absence of conditional effects this is the effect united with the precondition on the variables unchanged by the effect. In the presence of conditional effects, this is insufficiently specified, because we do not know which effect conditions could be satisfied. We extend the *post*-function such that it requires an additional argument, namely the abstract state $A = A_1 \times \cdots \times A_n$ on which $o$ is applied:

$$post(A, o) = \underset{i=1}{\overset{n}{\times}} \bigcup_{a \in prior(A_i, o)} \begin{cases} \{b\} & \text{if } \langle V_i, a, b \rangle \in eff(o) \\ \{a\} & \text{otherwise} \end{cases}$$

$$\tag{2}$$

where $\times$ denotes the generalized Cartesian product and $prior(A_i, o)$ captures which values of $A_i$ allow applying $o$. More formally, $prior(A_i, o)$ is defined as follows:

$$prior(A_i, o) = \begin{cases} \emptyset & \text{if } V_i \in pre(o) \wedge pre(o)[V_i] \notin A_i \\ \{a\} & \text{if } V_i \in pre(o) \wedge \\ & a = pre(o)[V_i] \in A_i \\ A_i & \text{otherwise (i.e., if } V_i \notin pre(o)). \end{cases} \tag{3}$$

Recall that operators are conflict-free and hence at most one effect $\langle V, v, v' \rangle$ exists for each atom $V \mapsto v$ in Equation 2. Clearly it holds that $post(A, o)[V] \subseteq \mathrm{dom}(V)$ for all $V \in \mathcal{V}$ which means $post(A, o)$ is a Cartesian set.

**Theorem 2.** *The Cartesian set $post(A, o)$ describes exactly the successor states of factored effect operator $o$ applied in any concrete state of the abstract stete $A$ where $o$ is applicable.*

*Proof.* We need to show two properties: (1) for all $s \in A$ with $pre(o) \subseteq s$ there is an $s' = s[\![o]\!] \in post(A, o)$; (2) for all $s' \in post(A, o)$ there is an $s \in A$ with $pre(o) \subseteq s$ and $s[\![o]\!] = s'$. Due to the characteristics of factored effect operators, we can again reason about each variable individually.

(1) Let $s \in A$ be a concrete state such that $pre(o) \subseteq s$ and let us consider variable $V_i$. If $V_i \in vars(pre(o))$, then $s[V_i] \in \{s[V_i]\} = prior(A_i, o)$ and if $V_i \notin vars(pre(o))$, then $s[V_i] \in A_i = prior(A_i, o)$.
We continue by rewritig the definition of the successor state $s' = s[\![o]\!]$ of Equation 1 for the special case of factored effect operators as follows:

$$s'[V] = \begin{cases} v & \text{if } \langle V, s[V], v \rangle \in \textit{eff}(o) \\ s[V] & \text{otherwise} \end{cases}$$

Since this corresponds to the case distinction in Equation 2 and because $s[V_i] \in prior(A_i, o)$ it follows that $s'[V_i] \in post(A, o)[V_i]$.
(2) Let $s' \in post(A, o)$. From Equation 2 it follows that $prior(A_i, o) \neq \emptyset$ for all variables. Hence, for each variable $V_i$ it must either hold that $V_i \in vars(pre(o))$ and $pre(o)[V_i] \in A_i$, or $V_i \notin pre(o)$.
In both cases, it directly follows that there exists a value $v \in A_i$ such that $s[V_i] = v$. For the former case, $v = pre(o)[V_i]$ and for the latter case $v$ can be chosen arbitrarily from $A_i$.
Note that it is not important whether there is an effect $\langle V_i, v, s'[V_i] \rangle \in \textit{eff}(o)$ because if there is an effect, then $s[V_i] = v$ due to the effect condition and otherwise $s[V_i] = s'[V_i] = v$.

$\square$

This leads to Algorithm 1 which is an adaption of Algorithm 5 by Seipp and Helmert (2018). While Seipp and Helmert distinguish three cases, we have only one case for two reasons. (1) Their first case is obsolete because our definition of $prior(A_i, o)$ in Equation 3 takes care of unsatisfied preconditions; setting $prior(A_i, o)$ to $\emptyset$ results in $post(A, o)[V_i] = \emptyset$ and trivially the cut with $B_i$ is empty in

---

**Algorithm 1:** Transition check. Returns true iff factored effect operator $o$ induces at least one transition between abstract states $A$ and $B$.

---
**1 function** CHECKTRANSITION($A, o, B$)
**2**    **for each** $V_i \in \mathcal{V}$ **do**
**3**       **if** $post(A, o)[V_i] \cap B_i = \emptyset$ **then**
**4**          **return** false
**5**    **return** true

---

Line 3 of Algorithm 1. (2) Their third case is obsolete because $vars(post(A, o)) = \mathcal{V}$.

## Rubik's Cube Model

Rubik's Cube is a permutation puzzle invented by Ernő Rubik in 1974 (see Figure 1). It is a 3-dimensional cube with 6 *faces*. The cube is sliced into 3 *layers* in each dimension, resulting in 27 smaller cubes called *cubies*. The faces of the cubies are called *facelets* and each cubie is colored differently on all its facelets. The puzzle is solved when all facelets on the same face of the cube have the same color. Each layer can be rotated by multiples of 90° around the cubie in its center. Rotating a layer rearranges the cubies and changes the colors on the faces. In an initial configuration of the puzzle, the cube is scrambled by an arbitrary, unknown sequence of rotations. The goal is to find a sequence of rotations which leads to the solved state where all faces show a single color. Simple algorithms exist which find such sequences (Kociemba 1992), but finding a shortest such sequence is challenging.

We describe some properties of Rubik's Cube in the following list:

(1) Rotating a layer changes the position and orientation of its cubies, but never reveals new facelets. The number and colors of visible facelets never changes for any cubie.

(2) There are 4 kinds of cubies differing by the number of visible facelets: 1 *inner* cubie not visible at all; 6 *center* cubies showing 1 facelet; 12 *edge* cubies showing 2 facelets; and 8 *corner* cubies showing 3 facelets.

(3) Rotating the middle layer of any dimension is equivalent to rotating the other layers of the same dimension in the opposite direction. Hence, we forbid rotating the middle layer without changing the possible configurations of Rubik's Cube. As a consequence, the center cubies never change their position and we can use them to identify the cube faces.

(4) There are 12 positions for the edge cubies and 8 positions for the corner cubies. Each position can be described using the faces it touches.

(5) At each position, edge cubies can occur in 2 possible orientations and corner cubies can occur in 3 possible orientations.

Based on these observations, we model Rubik's Cube as a family of factored effect tasks $\Pi_i = \langle \mathcal{V}_e \cup \mathcal{V}_c, \mathcal{O}, s_i, s_\star \rangle$. We address the faces as front ($F$), back ($B$), left ($L$), right ($R$), up ($U$), and down ($D$). $\mathcal{F}_i$ denotes the faces which lie

on the $i$-th dimension of Rubik's Cube: $\mathcal{F}_1 = \{F, B\}$, $\mathcal{F}_2 = \{L, R\}$, $\mathcal{F}_3 = \{U, D\}$. Furthermore, $\mathcal{F} = \bigcup_i \mathcal{F}_i$ is the set of all faces and the faces *adjacent* to some $f \in \mathcal{F}_i$ is $\mathcal{A}_f = \mathcal{F} \setminus \mathcal{F}_i$. We denote the face that follows face $f'$ in clockwise turn order from the perspective of face $f$ as $\mathcal{A}_f[f']$, e.g., $\mathcal{A}_F[U] = R$.

$\mathcal{V}_c$ contains 8 variables representing the corner cubies. According to observations (4) and (5) they can be described by their position using the three faces they touch and by one of three possible orientations. Hence, the domain of a corner cubie $V_c \in \mathcal{V}_c$ is $\mathrm{dom}(V_c) = \{\langle \{f_1, f_2, f_3\}, o \rangle \mid f_1 \in \mathcal{F}_1, f_2 \in \mathcal{F}_2, f_3 \in \mathcal{F}_3, o \in \{1, 2, 3\}\}$. Similarly, $\mathcal{V}_e$ contains 12 variables representing the edge cubies. To describe the edge cubies, we require their position described by the two faces they touch and their orientation. Hence, the domain of an edge cubie $V_e \in \mathcal{V}_e$ is $\mathrm{dom}(V_e) = \{\langle \{f_1, f_2\}, o \rangle \mid f_1 \in \mathcal{F}_i, f_2 \in \mathcal{F}_j, i \neq j, o \in \{1, 2\}\}$.

$\mathcal{O}$ has one operator *rotate*$(f, angle)$ for each face $f \in \mathcal{F}$ and for the angles 90° (clockwise), 180° (half-turn), and 270° (counter-clockwise). All operators $o \in \mathcal{O}$ are always applicable, i.e., $pre(o) = \emptyset$. An operator *rotate*$(f, angle)$ changes the position and rotation of all cubies on $f$.

We start by describing the effect of *rotate*$(f_1, 90)$. We first model how the cubies on face $f_1$ change their position. A corner cubie at position $\{f_1, f_2, f_3\}$ moves to $\{f_1, \mathcal{A}_{f_1}[f_2], \mathcal{A}_{f_1}[f_3]\}$. More specifically, the faces $f_2$ and $f_3$ are updated to the next face in clockwise direction (from the perspective of $f_1$). Similarly, an edge cubie at position $\{f_1, f_2\}$ moves to position $\{f_1, \mathcal{A}_{f_1}[f_2]\}$.

To easily model how the orientation changes, we use a trick. We represent the orientation of a cubie as a triple. More specifically, we represent the orientation of a corner cubie as a permutation of $\{1, 2, 3\}$, and the orientation of an edge cubie as a permutation of $\{1, 2, \#\}$ where $\#$ represents a *blank* symbol. The first non-blank value identifies the orientation of the cubie. If we now rotate any cubie with an orientation $\langle o_1, o_2, o_3 \rangle$ around the dimension $d \in \{1, 2, 3\}$, then the new orientation of the cubie is computed by exchanging the elements of the triple which are not at position $d$. For example, a rotation around $d = 3$ leads to the new triple $\langle o_2, o_1, o_3 \rangle$. To see why this is possible, observe that for any corner cubie, there are only three different triples reachable at every position and for any edge cubie there are only 2 triples reachable at any position. Thus, given the cubie, its position, and its orientation, we can construct the correct triple, modify it, and extract the new orientation value. We call the function described in this paragraph *next_orientation*.

Putting everything together, the effects of a 90° rotation of $f_1$ are:

$$eff(rotate(f_1, 90)) = \{$$
$$\langle V_c, \langle \{f_1, f_2, f_3\}, o \rangle, \langle \{f_1, \mathcal{A}_{f_1}[f_2], \mathcal{A}_{f_1}[f_3]\}, o' \rangle \rangle \mid$$
$$V_c \in \mathcal{V}_c, f_1 \in \mathcal{F}_d,$$
$$o' = next\_orientation(V_c, \{f_1, f_2, f_3\}, o, d)$$
$$\} \cup \{$$
$$\langle V_e, \langle \{f_1, f_2\}, o \rangle, \langle \{f_1, \mathcal{A}_{f_1}[f_2]\}, o' \rangle \rangle \mid$$
$$V_e \in \mathcal{V}_e, f_1 \in \mathcal{F}_d$$
$$o' = next\_orientation(V_c, \{f_1, f_2\}, o, d)$$
$$\}$$

Rotating a face by 180° (respectively 270°) is modeled by replacing every occurrence of $\mathcal{A}_{f_1}[\cdot]$ by two (respectively three), calls to the same function, i.e., $\mathcal{A}_{f_1}[\mathcal{A}_{f_1}[\cdot]]$ (respectively $\mathcal{A}_{f_1}[\mathcal{A}_{f_1}[\mathcal{A}_{f_1}[\cdot]]]$), and doing the same for *next_orientation*.

The goal state $s_\star$ has a single cubie in each position and all cubies have an orientation of 1. Any state reachable by applying a sequence of operators can be the initial state $s_I$.

## Experiments

We evaluate the performance of heuristics based on PDBs, Cartesian abstractions, and M&S abstractions on Rubik's Cube problems using the Fast Downward planning system (Helmert 2006). We add support for factored effect tasks in the generation of Cartesian abstractions using CEGAR. In this section, we describe the tasks we use for our evaluation, give an overview of the compared planner configurations (i.e., heuristics), and evaluate their performance. Our code, benchmarks, and data are available online (Büchner et al. 2022).

### Benchmarks

We implement a SAS$^+$ problem generator for Rubik's Cube tasks according to our model. Our generator starts from the solved configuration and applies $n$ arbitrary rotations where $n$ is a user specified value. We ensure that two consecutive rotations concern different faces, because such rotations can be combined into one rotation. Nevertheless, some sequences of rotations can cancel each other out. Thus, $n$ is just an upper bound on the minimum plan length.

We generate 10 tasks for each $n \in \{1, \ldots, 20\}$ to obtain a benchmark set with diverse difficulty levels. We choose 20 as the upper limit because Rokicki et al. (2014) show that all Rubik's Cube instances can be solved with at most 20 moves. We ensure the benchmark set contains no duplicates, which leaves us with a total of 200 distinct Rubik's Cube instances for our evaluation.

### Evaluated Heuristics

While we compare the performance of three abstraction classes, there is a total of five configurations that we include in our evaluation. We briefly describe them here.

**Blind Heuristic** We use a blind search (denoted as BLIND) as a baseline for our investigations. It corresponds to a breadth-first exploration of the transition system.

**Pattern Database Heuristics** For PDBs, the crucial decisions include how to choose the patterns and how to combine their individual heuristic values. Korf (1997) split the variables into two patterns, one for the edge and one for the corner cubies. Korf used a specialized solver explicitly designed for Rubik's Cube. In contrast, Fast Downward is a domain-independent planner without optimizations for Rubik's Cube. Thus, it cannot fit Korf's PDBs into memory and we are not able to compare to this state-of-the-art approach within this experiment. Instead, we use the largest patterns which fit into memory for our Rubik's Cube model. Those consist of 4 variables. We use the following two configurations for PDBs:

PDB-MAN Inspired by Korf's approach, we have two patterns containing the corner cubies: one for the cubies on the front face and one for the cubies on the back face. Additionally, we have three patterns for the edge cubies: one for the cubies on the front face, one for the cubies on the back face, and one for the cubies between the front and back face.

PDB-SYS This configuration systematically generates all *interesting* patterns up to a certain size (Pommerening, Röger, and Helmert 2013). For Rubik's Cube, this strategy turned out to be feasible for patterns of size up to 3.

In both cases, the heuristic is the maximum over all these patterns.

**Cartesian Abstraction Heuristic** By CEGAR we denote the Cartesian CEGAR algorithm as discussed in this paper. We do not limit the number of states or transitions in the abstraction. The abstraction refinement is terminated to start the search after at most 900 seconds or if the memory limit is approached.

**Merge-and-Shrink Heuristic** The last configuration we consider is M&S. It uses bisimulation as shrinking strategy (Nissim, Hoffmann, and Helmert 2011), strongly connected components as merging strategy (Sievers, Wehrle, and Helmert 2016), and exact label reduction (Sievers, Wehrle, and Helmert 2014). We limit the abstraction to at most 50,000 states.

## Setup

We implement our extensions in the Scorpion planner (Seipp, Keller, and Helmert 2020), an extension of Fast Downward (Helmert 2006). Scorpion already contains implementations of PDBs for tasks with conditional effects as well as a version of Cartesian CEGAR using incremental search (Seipp, von Allmen, and Helmert 2020). We use Lab (Seipp et al. 2017) for running our experiments. We execute $A^*$ searches with each of the five heuristics from above on all 200 benchmark tasks. The experiment is conducted on Intel Xeon-Silver 4114 processors running on 2.2 GHz with a time limit of 30 minutes and a memory limit of 3.5 GB.

## Results

The most successful heuristic in terms of solved tasks is PDB-MAN which solves 123 tasks. PDB-SYS is almost as successful with 119 solved tasks, followed by CEGAR which solves 113 tasks. M&S falls behind with only 90 solved tasks and BLIND solves, as expected, the fewest tasks, namely 66. For PDB-SYS, all failed runs are due to the time limit. For all other configurations, the failed runs are due to the memory limit.

None of the heuristics is able to solve problems where the optimal solution requires more than 13 rotations. According to Rokicki et al. (2014), this restriction allows us to solve only 0.0001% out of the $4.3 \cdot 10^{19}$ possible states of Rubik's Cube. In comparison, Korf (1997) was able to optimally solve tasks at least 18 rotations away from the goal.[1] The set of initial states with optimal cost 18 or less makes up approximately 98% of all possible initial states for Rubik's Cube (Rokicki et al. 2014).

Figure 2 plots the number of states expanded before the last $f$-layer of $A^*$.[2] We plot against PDB-MAN which is the heuristic that solves the most problems. The figure reveals that both PDB variants are similar in terms of heuristic guidance, as they expand similarly many states. M&S generally expands more states. This indicates that M&S abstractions (using our parameters) capture less of the essence of Rubik's Cube than the simpler projections on manually chosen variables. This is surprising given that M&S is the more general class of abstractions. However, an important difference is that PDBs make use of multiple abstractions and use the highest heuristic estimate among all of them while M&S constructs a single abstraction.

In contrast, CEGAR seems to produce very fine-grained abstractions up to a certain size; the data points on the bottom line indicate that we obtain perfect heuristic values before the last $f$-layer from our Cartesian abstractions. However, this changes abruptly when the problem instances become harder. At some point, the guidance quickly deteriorates and becomes worse than the guidance of PDB-MAN.

Looking at the runtimes reveals more information about the techniques for generating abstractions in the first place. Figure 3 plots the runtime of all configurations against CEGAR because it has the most evenly distributed runtimes for the problems of different difficulties. BLIND is faster on the trivial tasks but is quickly outperformed by all other heuristics. For PDB-MAN, PDB-SYS, and M&S, the preprocessing time (approximately 10s, 77s, and 17s, respectively) required for constructing the abstractions and precomputing the heuristic dominates the actual search time on most instances. Only after a certain difficulty, the search time becomes visible in Figure 3, namely when the constant line starts to ascend. Meanwhile, CEGAR allows to terminate the construction of the abstraction early if it finds a solution

---

[1]Back in 1997, it took around four weeks to solve such problems.

[2]We ignore the expansions in the last $f$-layer, because all states in this layer, including the goal state, have the same priority for $A^*$ and it is due to random tie-breaking in which order they are expanded.

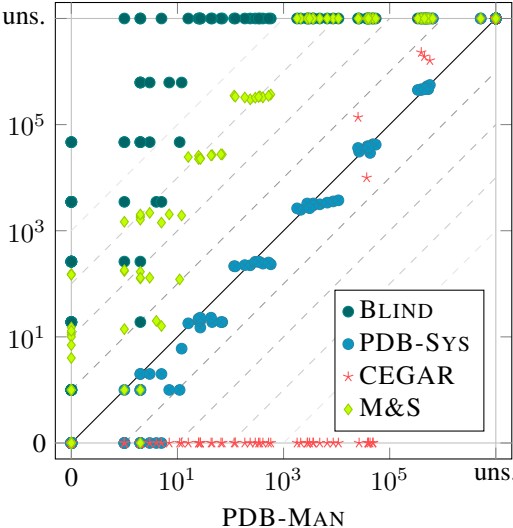

Figure 2: Number of state expansions before the last $f$-layer.

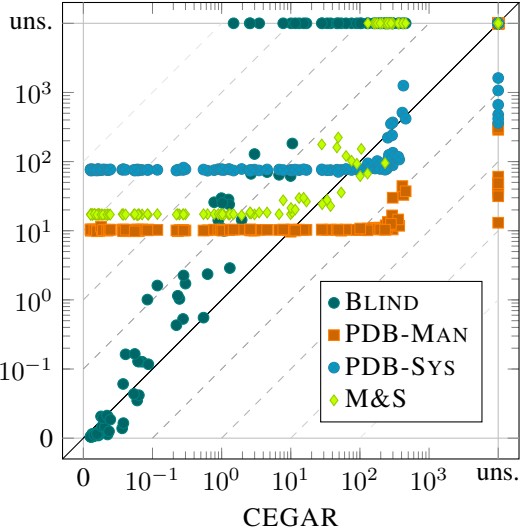

Figure 3: Runtime in seconds.

during the refinement step. This is an advantage that pays off for the simpler problems but is later outmatched by the PDB based techniques.

## Discussion

Overall, the performance of CEGAR looks promising and competitive until we reach the point where the heuristic quality deteriorates and many state expansions are required. Since the state space of Rubik's Cube is huge and the solution tracing always starts from the initial state, we assume that the abstraction is fine-grained locally around the initial state, but becomes almost blind when reaching the borders of that area. This is in contrast to the pattern databases, which ignore parts of the information evenly across the entire state space.

The poor performance of M&S is another surprising result. The initial heuristic values for M&S are the worst among all considered heuristics and they never surpass the value 4. Potentially, the chosen size limit of at most 50'000 abstract states is too restrictive for Rubik's Cube.

## Conclusion

We analyzed the performance of modern abstraction heuristics on the famous Rubik's Cube problem. First, we presented a model for Rubik's Cube as a finite-domain planning task which allows us to use general problem solvers. Our model relies on a special kind of conditional effect where the effect condition is concerned only with the variable changed by the effect; when rotating a face of Rubik's Cube, the cubies change location and orientation based on their own previous location and orientation. We introduced these problems as factored effect tasks.

While conditional effects are supported in PDBs and M&S abstractions, this is not the case for Cartesian abstractions created using counterexample-guided Cartesian abstraction refinement. Hence, we extended this theory for factored effect tasks.

Finally, we evaluated the performance of heuristic search based on the aforementioned abstraction classes on a newly generated benchmark set of Rubik's Cube tasks. The experiment reveals that PDBs remain the state-of-the-art for this domain, at least in terms of overall coverage. However, Cartesian CEGAR yields strong heuristics for problem instances up to a certain difficulty. Future work should investigate further what exactly the limitation is for Cartesian CEGAR.

A missing piece in the hierarchy of abstractions are *domain abstractions*. We are not aware of any work that generates domain abstractions for classical planning problems in a principled way. Filling this gap would be a major accomplishment and could shed more light on the analysis of abstraction heuristics for Rubik's Cube.

Rubik's Cube is one domain in a family of permutation puzzles with many similarities. Other domains like sliding tiles, the pancake problem, genome rearrangement, and many more can be modeled as factored effect tasks as well. It will be interesting to see whether the abstraction classes compare similarly for all of these domains.

## Acknowledgments

This research was supported by TAILOR, a project funded by the EU Horizon 2020 research and innovation programme (grant agreement no. 952215) and by the European Research Council (ERC) under the European Union's Horizon 2020 research and innovation programme (grant agreement no. 817639). Patrick Ferber was partially supported by DFG grant 389792660 as part of TRR 248 (see https://perspicuous-computing.science) and Jendrik Seipp was partially supported by the Wallenberg AI, Autonomous Systems and Software Program (WASP) funded by the Knut and Alice Wallenberg Foundation.

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
