# OpenReview forum: "A Comparison of Abstraction Heuristics for Rubik's Cube"
_icaps-conference.org/ICAPS/2022/Workshop/HSDIP — HSDIP 2022_

### Official Review · Reviewer_EAyb · 2022-04-24
**Planning solution to Rubrik's; odd performance outcomes**

**Confidence:** 4
**Overall Score:** Strong Accept

**Review:**

This paper introduces a planning model and evaluation of the common rubik's cube puzzle. It was a joy to read, and it's great to see this setting finally modelled for planners.

Seeing the results summarized early in the paper, it was confusing how counter-intuitive the outcome actually is (with approaches effectively flipped in what would be expected). I think it may be worth acknowledging this odd behaviour early in the paper, and point out that you discuss it further (along with some conjectures) later on. The conjecture on CEGAR being focused locally around the initial state seems plausible, and it may be something worth investigating -- for smaller instances, compare the quality of the heuristic as a function of distance from the initial state. Also, if this is truly the case, it's interesting to consider how it might be remedied -- i.e., how CEGAR could be reworked so that the quality of heuristic guidance is not front-loaded.

One low-hanging fruit for this analysis would be to try on the other domains identified as having factored effects. From what I can tell, this would be an easy thing to do (given that the methods are already implemented) and would inform us on if the observed phenomenon is likely due to the factored effects or Rubik's specifically.

One final concern I have is with the rhetoric around factored effects. At times, it felt like the paper indicated that specialized solvers were required for this setting. From what I understand, it's a specific case of the more general FDR, and the restriction exists only to allow a generalization of CEGAR to condifitional effects. Using other heuristics/planners on the model would help convey that it is indeed in a form agnostic to the planner used.

In the end, I think this paper would be a welcome addition to the HSDIP program.

## Minor fixes
- "Rotating a face for 180° or 270° is modeled by replacing every occurence of Af1[f2] by two respectively 3 calls" > "...180° (resp. 270°)...occur**r**ence...two (resp. three) calls"
- "50’000" > "50,000"
- "CEGAR yields constructs regularly the perfect heuristic" > remove yields

---

### Official Review · Reviewer_VWbY · 2022-04-25
**Good paper on CEGAR with conditional effects**

**Confidence:** 4
**Overall Score:** Accept

**Review:**

The paper deals with the application of cartesian abstractions on the problem of Rubik's cube formulated as a planning problem with (certain subclass of) conditional effects.

The paper is interesting, easy to read and it fits HSDIP.

I have just few suggestions for improvements:
1. The formulation "special kind of conditional effects" sounded a little bit misleading to me (so that it actually downplays the contribution of the paper). From the beggining, I assumed that the authors propose some kind of a new type of conditional effects, only to find out on the third page that they use a certain subclass of standard conditional effects, which, in my opinion, are actually quite common. So, I would suggest to use a clearer formulation and decribe what type of conditional effects are used right from the beggining -- so basically spell out (informally) Def. 1 in the introduction (or even the abstract).

2. The focus on Rubik's cube, I think, downplays the contribution of the paper, because what authors actually show is how to use CEGAR with what they call "factored effect operators", which is a significant contribution in its own right. So, it's worth considering whether it wouldn't be better to present the material as primarly focused on CEGAR for a certain subclass of conditional effects and use Rubik's cube as a case-study. But this would, probably, be too big change to HSDIP camera-ready.

3. The theoretical parton page 3 is, in my opinion, a little bit sloppy -- Theorem 1 is not clearly formulated because it doesn't say what exactly is "regression of a Cartesian set with respect to a factored effect operator". It is spelled out only in the proof. So, I would suggest to first define the regression (that's basically the equation in the proof of Theorem 1) and then formulate Theorem 1. Another problem is that Theorem 1 is missing the last step of the reasoning -- it proves the claim for a single variable and then it simply states that this extends to all variables without actually proving this step. Separating the definition of regression and the Theorem will, I think, make it clearer this part of the argument is missing.

Minor issues:

Def. 1: it's never stated where v and v' come from -- I think just adding "for some v, v' \in V" at the end of the first sentence would suffice.

The paragraph under Def. 2 states additional condition on the factored effect operators to ensure that they are conflict-free. Why isn't this part of Def. 1? I think it would make the formal part a little bit clearer (in fact, the whole paragraph under Def. 2 can be part of Def. 1). You can then call these operators "Conflict-free factored effect operators" if you want to distinguish them from the ones with conflicts, but I don't think it's necessary, because conflict-free property is very reasonable because otherwise the semantics of conditional effects may not be clear.

---

### Author Response · Authors · 2022-04-27
**Response to Reviewers**

We thank the reviewers for their insightful feedback.

We will fix the minor issues pointed to us before the camera-ready copy
and address the other feedback where possible.


It seems that in the beginning, the necessary restrictions to extend
CEGAR to conditional effects comes off too strong. We will try to relax
and clarify them.

We are not entirely sure how to interpret the suggestion in the first
review to use other heuristics/planners to help convey our model does
not need a specialized planner. We believe we already do so by comparing
to blind search, PDBs, and merge-and-shrink, as no adaptions of Fast
Downward are needed to evaluate these methods.

Indeed, Rubik's Cube is mainly a case study for our extension of the
CEGAR theory to conditional effects. In the future, we plan to implement
problem generators for other permutation domains that fit the
requirements for factored effect tasks. Our plans therefore go in the
direction proposed by the second review, eventually moving away from the
focus on Rubik's Cube and focusing more on theoretical aspects of the
abstraction classes for such problems. However, we also see the Rubik's
Cube model as an important contribution, and we agree that changing this
for the camera-ready copy of the workshop would be quite a big change,
so we stick with this presentation for now.


Thanks again for the constructive suggestions.